# Risk Factors for Problematic Drinking in One’s Thirties and Forties: A Longitudinal Analysis of the 1970 British Cohort Study

**DOI:** 10.3390/ijerph191710664

**Published:** 2022-08-26

**Authors:** Gary O’Donovan, Mark Hamer

**Affiliations:** 1Latin American Brain Health Institute (BrainLat), Universidad Adolfo Ibáñez, Peñalolén, Santiago 7941169, Chile; 2Facultad de Medicina, Universidad de los Andes, Carrera 1, 18A-12, Bogotá 111711, Colombia; 3Division of Surgery & Interventional Science, Faculty of Medical Sciences, Institute Sport Exercise Health, University College London, 170 Tottenham Court Road, London WC1E 6BT, UK

**Keywords:** alcohol drinking, alcohol-related disorders, risk factors, cohort studies, primary prevention, secondary prevention

## Abstract

Alcohol drinking and risk factors for problematic drinking may vary across a lifespan. The objective of this study was to identify risk factors for problematic drinking in men and women in their thirties and forties. Alcohol drinking and potential risk factors for problematic drinking were assessed at ages 30, 34, 42, and 46 in the 1970 British Cohort Study. Multilevel models included 10,079 observations in 3880 men and 9241 observations in 3716 women. In men, formerly smoking, currently smoking, having a degree, having malaise, and having a mother who drank while pregnant were independently associated with increased risk of problematic drinking. In women, formerly smoking, currently smoking, being physically active in one’s leisure time, having a degree, having a managerial or professional occupation, having malaise, and having a mother who drank while pregnant were independently associated with increased risk of problematic drinking. In men and women, cohabiting as a couple was associated with decreased risk of problematic drinking. This study suggests that several risk factors may be associated with problematic drinking in men and women in their thirties and forties. Policy makers should consider the role of modifiable risk factors in the prevention of problematic drinking.

## 1. Introduction

Alcohol use and alcohol use disorder are among the leading causes of death and disability worldwide [1]. In England alone, 45,210 men and 30,345 women sought treatment from the state for alcohol problems between 1 April 2018 and 31 March 2019; A further 28,598 adults sought treatment for non-opiate drug and alcohol problems [2]. The cost of adult drug services in England is around £481 million per year, with a further £222 million being spent on adult alcohol services [3]. Data from England and elsewhere in the United Kingdom suggest that alcohol consumption varies across the lifespan: there is a rapid increase in alcohol consumption during adolescence, a plateau in midlife, and a decline into older age [4,5]. Risk factors for problematic alcohol drinking may also vary across the lifespan [6]. For example, in early middle-aged adults, adverse childhood experiences and parental alcohol use may be associated with higher alcohol drinking and being married may be associated with lower alcohol drinking [6]. Alcohol drinking is only measured at baseline in most prospective cohort studies and more longitudinal research with repeated measurements is required to understand complex associations between potential risk factors and problematic drinking [5,6]. The 1970 British Cohort study is a longitudinal study with repeated measurements and the objective of the present study was to identify risk factors for problematic drinking in cohort members in their thirties and forties. Most studies about risk factors for problematic drinking are cross-sectional and, to the best of our knowledge, this longitudinal study is the first study to include repeated measurements of alcohol drinking and potential risk factors for problematic drinking [6].

## 2. Materials and Methods

### 2.1. Participants

The 1970 British Cohort Study consists of people born in England, Scotland and Wales during a single week in 1970 [7,8]. The present analysis included data from the age 10 survey (1980–1981), age 30 survey (1999–2000), age 34 survey (2004–2005), age 42 survey (2012–2013), and age 46 survey (2016–2018) [9,10,11,12]. At the age 10 survey, health visitors went to the homes of cohort members and conducted interviews with parents [9]. Parents and cohort members were also asked to complete questionnaires [9]. At the age 30 and age 34 surveys, health visitors went to the homes of cohort members and conducted interviews with cohort members and asked them to complete questionnaires [10,13]. At the age 42 and age 46 surveys, health visitors and/or nurses went to the homes of cohort members and conducted interviews with cohort members and asked them to complete questionnaires [11]. The interviewer transcripts and the self-completion questionnaires used in the 1970 British Cohort Study are available online [8,14]. Parents (age 10 survey) and participants (age 30, age 34 and age 46 surveys) provided informed consent. The ethical review procedures used in the 1970 British Cohort Study are described in detail on the study website [15]. The age 10 survey was subject to internal review only, the age 30 survey was approved by the London Multicentre Research Ethics Committee (98/2/120), the age 34 survey was subject to internal review only, the age 42 survey was approved by the London–Central research ethics committee (11/LO/1560), and the age 46 survey was approved by the South East Coast–Brighton and Sussex research ethics committee (15/LO/1446).

### 2.2. Dependent Variable

The dependent variable was problematic drinking. At ages 30 and 34, the cutting down, being annoyed by criticism, feeling guilty, and eye-openers (CAGE) questionnaire was used to assess problematic drinking [16]. Problematic drinking was defined as two or more affirmative replies to four questions [16]: Have you ever felt that you ought to cut down on your drinking? Have people annoyed you by criticizing your drinking? Have you ever felt bad or guilty about your drinking? Have you ever had a drink first thing in the morning to steady your nerves or get rid of a hangover? The CAGE questionnaire is regarded as a valid screening tool in general practice [16]. At ages 42 and 46, cohort members were asked the five questions that make up the Alcohol Use Disorders Identification Test for Primary Care (AUDIT-PC): how often do you have a drink containing alcohol? (never scores 0; monthly or less scores 1; two to four times a month scores 2; two to three times a week scores 3; four or more times a week scores 4); how many drinks containing alcohol do you have on a typical day when you are drinking? (one to two scores 0; three to four scores 1; five to six scores 2; seven to nine scores 3; ten or more scores 4); how often in the last year have you found that you were not able to stop drinking once you started? (never scores 0; less than monthly scores 1; monthly scores 2; weekly scores 3; daily or almost daily scores 4); how often during the last year have you failed to do what was normally expected from you because of your drinking? (never scores 0; less than monthly scores 1; monthly scores 2; weekly scores 3; daily or almost daily scores 4); has a relative or friend, doctor or other health worker been concerned about your drinking and suggested that you cut down? (no scores 0; yes, but not in the last year scores 2; yes, during the last year scores 4). Total AUDIT-PC scores of 0–4 were considered unproblematic drinking and total scores of five or more were considered problematic drinking. The ten-question alcohol use disorders identification test and shorter versions are regarded as valid screening tools for the detection of alcohol use disorder in the general population when compared with the criterion measure, the Diagnostic and Statistical Manual of Mental Disorders [17,18,19]. Face to face computer aided personal interviewing (CAPI) is the preferred means of data collection in the 1970 British Cohort Study. Indeed, alcohol drinking was assessed using CAPI interviews at age 30, age 34, and age 46. However, alcohol drinking was assessed using paper self-completion questionnaires at age 42. A total of 8600 of 9692 (89%) paper questionnaires were completed at age 42 [12].

### 2.3. Independent Variables

The independent variables included six potentially modifiable risk factors for problematic drinking, which were assessed at age 30, age 34, age 42, and age 46: smoking, leisure-time physical activity, highest academic qualification, occupation, cohabiting as a couple, and malaise [2,20,21,22,23,24]. The independent variables also included two non-modifiable risk factors for problematic drinking, which were assessed at age 10: whether the cohort member’s mother drank while pregnant and the cohort member’s father’s occupation in the year of the cohort member’s birth [25,26]. Cohort members were asked about their smoking habit and three categories were derived: never smoked, former smoker, and current smoker. Cohort members were asked about their leisure-time physical activity habit and four categories were derived: none; once a week or less; two or three times a week; and, four or more times a week. Cohort members were asked about their highest academic qualification and four categories were derived: none; GCSE or equivalent; A-level or equivalent; and, degree or higher degree. Cohort members were asked about their occupation and three categories were derived: unskilled; semi-skilled or skilled; and, managerial or professional. Cohort members were asked whether or not they were cohabiting as a couple. Cohort members were asked to complete the 24-item Malaise Inventory at age 30 and a nine-item version at age 34, age 42, and age 46 [27]. A malaise score of eight or more was regarded as high at age 30 and a score of four or more was regarded as high at age 34, age 42, and age 46. The cohort member’s mother was asked whether or not she took an alcoholic drink during her pregnancy. The cohort member’s father was asked about their occupation and three categories were derived: unskilled; semi-skilled or skilled; and, managerial or professional.

### 2.4. Statistical Methods

Repeated measurements of the same individual are highly correlated, so we used the multilevel models that can cope with the problem of correlated observations in longitudinal studies [28]. All analyses were performed using Stata MP version 15.1 for Mac (StataCorp, College Station, TX, USA). The *melogit* command was used to fit multilevel logistic regression models for the binary outcome, problematic drinking. Models were fitted for men and women separately because alcohol drinking trajectories are different in men and women in the United Kingdom [5]. We used linear models that allowed for random intercepts. Likelihood-ratio tests comparing each model with ordinary logistic regression were highly significant (all *p* < 0.001). Problematic drinking, smoking, leisure-time physical activity, highest academic qualification, occupation, cohabiting as a couple, and malaise were time varying variables and all the available data from each wave were used in each model. Mother drinking during pregnancy and father’s occupation were unvarying and the data from the age 10 survey were used in each model. All variables were treated as categorical variables. Few cohort members and few cohort members’ fathers were in unskilled occupations, so the unskilled category and the semi-skilled or skilled category were combined. Multilevel logistic regression models allow for participants to be problematic drinkers at one wave and not another. Nonetheless, we excluded participants with problematic drinking at age 30 in sensitivity analyses because it was thought that they might be more susceptible to problematic drinking at age 34, age 42, or age 46.

## 3. Results

Appendix A shows the flow of participants through the study. Cohort members were not included in a given wave in the present analysis if they were missing data for the dependent variable and the independent variables. Appendix A shows problematic drinking frequency in cohort members who were and were not included in the present analysis. The frequency was similar in those who were and were not included at every wave. For example, problematic drinking frequency was 13.96% in those who were and 13.80% in those who were not included at age 30; Additionally, the frequency was 23.93% in those who were and 23.05% in those who were not included at age 46. All the available data were used in the present analysis, whether from the minimum of one wave or the maximum of four waves. Table 1 shows male participants’ characteristics. Around 20% of men screened positive for problematic drinking in their thirties according to the CAGE questionnaire and more than 30% screened positive in their forties according to the AUDIT-PC questionnaire. The proportion of men with a degree or higher degree, the proportion of men in managerial or professional occupations, the proportion of men cohabiting as a couple, and the proportion of men with a high malaise score increased with age. The proportion of men who smoked decreased with age. Around 20% of men reported no leisure-time physical activity at every wave. Around 50% of the men’s mothers drank during pregnancy and around 30% of the men’s fathers were in managerial or professional occupations. Table 2 shows female participants’ characteristics. Less than 15% of women screened positive for problematic drinking in their thirties and less than 20% in their forties. The proportion of women with a degree or higher degree, the proportion of women in managerial or professional occupations, the proportion of women cohabiting as a couple, and the proportion of women with a high malaise score increased somewhat with age. The proportion of women who smoked decreased with age. Around 20% of women reported no leisure-time physical activity in their thirties and more than 25% reported no leisure-time physical activity in their forties. Around 50% of the women’s mothers drank during pregnancy and around 30% of the women’s fathers were in managerial or professional occupations.

Table 3 shows longitudinal associations of risk factors with problematic drinking in men. The multilevel model included 10,079 observations in 3880 men. Some potentially modifiable risk factors were associated with problematic drinking. Formerly smoking (3.06; 2.52, 3.72) and currently smoking (3.76; 3.09, 4.57) were particularly strong predictors of problematic drinking (values are odds ratios and 95% confidence intervals). Having a degree or higher degree and having a high malaise score were also independently associated with increased risk of problematic drinking. Cohabiting as a couple was associated with decreased risk of problematic drinking. The fact that one’s mother drank during pregnancy is not modifiable and was strongly associated with increased risk of problematic drinking in men in the present study (1.70; 1.43, 2.02). Appendix A shows longitudinal associations of risk factors with problematic drinking, excluding men with problematic drinking at age 30. The results of the sensitivity analysis were similar to the main analysis, but having a degree or higher degree was no longer associated with problematic drinking.

Potential risk factors were assessed at age 30, age 34, age 42, and age 46. The number of cases of problematic drinking at each age is reported in Table 1. A multilevel model was fitted to the data, which was a linear model that allowed for random intercepts. All variables in the model were time varying except mother drank during pregnancy and father’s occupation in 1970. The model included 10,079 observations in 3880 male cohort members. The average number of observations per cohort member was 2.6, where the minimum was 1 and the maximum was 4. Values are mutually adjusted odds ratios. GCSE is general certificate of education, a qualification usually sought around 16 years of age. A-level is advance level, a qualification usually sought around 18 years of age.

Table 4 shows longitudinal associations of risk factors with problematic drinking in women. The multilevel model included 9241 observations in 3716 women. Some modifiable and non-modifiable risk factors were associated with problematic drinking. Formerly smoking (2.74; 2.16, 3.48), currently smoking (4.92; 3.83, 6.31), having a degree or higher degree (2.01; 1.43, 281), and having a high malaise score (1.95; 1.56, 2.43) were particularly strong predictors of problematic drinking. Being physically active in one’s leisure time, having a managerial or professional occupation, and having a mother who drank while pregnant were also independently associated with increased risk of problematic drinking. Cohabiting as a couple was associated with decreased risk of problematic drinking. Appendix A shows longitudinal associations of risk factors with problematic drinking, excluding women with problematic drinking at age 30. The results of the sensitivity analysis were similar to the main analysis, but the association of physical activity with problematic drinking was no longer statistically significant.

Potential risk factors were assessed at age 30, age 34, age 42, and age 46. The number of cases of problematic drinking at each age is reported in Table 2. A multilevel model was fitted to the data, which was a linear model that allowed for random intercepts. All variables in the model were time varying except mother drank during pregnancy and father’s occupation in 1970. The model included 9241 observations in 3716 female cohort members. The average number of observations per cohort member was 2.5, where the minimum was 1 and the maximum was 4. Values are mutually adjusted odds ratios. GCSE is general certificate of education, a qualification usually sought around 16 years of age. A-level is advance level, a qualification usually sought around 18 years of age.

## 4. Discussion

The objective of the present study was to identify risk factors for problematic drinking in men and women in their thirties and forties. Alcohol drinking and potential risk factors for problematic drinking were assessed during four waves of the 1970 British Cohort Study and sophisticated analyses were used to cope with the problem of correlated observations. In men, formerly smoking, currently smoking, having a degree, having malaise, and having a mother who drank while pregnant were independently associated with increased risk of problematic drinking. In women, formerly smoking, currently smoking, being physically active in one’s leisure time, having a degree, having a managerial or professional occupation, having malaise, and having a mother who drank while pregnant were independently associated with increased risk of problematic drinking. In men and women, cohabiting as a couple was associated with decreased risk of problematic drinking. These findings have important implications for the primary prevention of problematic drinking.

Alcohol drinking is only measured at baseline in most prospective cohort studies and it has been suggested that such studies be treated with caution because drinking behavior changes with age [5]. Britton and colleagues [5] investigated drinking behavior in nine prospective cohort studies with at least three repeated measurements and they used multilevel models to cope with the problem of correlated observations. The models included 1,774,666 observations in 59,397 people in the United Kingdom and it was found that there was a rapid increase in the volume of alcohol consumed during adolescence, a plateau in midlife, and a decline into older age. Potential risk factors for problematic drinking have been suggested [6], but, to the best of our knowledge, the present study is the only prospective cohort study to include repeated measurements of alcohol drinking and potential risk factors for problematic drinking. The results of the main analysis and the sensitivity analysis were similar, demonstrating the robustness of the assessment. The participants with problematic drinking at age 30 who were excluded from the sensitivity analysis were more susceptible to problematic drinking, but reoccurrence was not inevitable: 57% were problematic drinkers at age 34, 50% at age 42, and 46% at age 46. These findings have important implications for both the primary and the secondary prevention of problematic drinking.

Nicotine activates stress- and reward-related brain regions that may facilitate the transition to compulsive alcohol drinking [29]; and, smokers in England are more likely to drink [2]. The present study suggests that formerly smoking and currently smoking are risk factors for problematic drinking in men and women in their thirties and forties. Physical activity may affect alcohol use by reducing cravings and improving mood [30]; however, cross-sectional data suggest that adults in England and Scotland who meet physical activity guidelines are more likely to drink frequently [20]. Cross-sectional data also suggest that women in the US with higher levels of aerobic fitness have higher levels of alcohol consumption [31]. The present prospective study also suggests that women who are physically active are more susceptible to problematic drinking. Cross-sectional studies suggest that highly educated adults in England are less likely to be problematic drinkers [21]. At the same time, such studies suggest that managers and other professionals are more likely to be problematic drinkers [21]. The largest cross-sectional study to date in the UK suggests that the risk of heavy alcohol consumption is particularly high in women who are directors and chief executives of major organizations [22]. The present prospective study also suggests that professionals are more susceptible to problematic drinking, particularly professional women in their thirties and forties. It is difficult to explain the observed associations between physical activity, professional occupations, and alcohol consumption, but some people drink to cope with long working hours [32] and some drink to reward themselves (the ‘work hard, play hard’ attitude) [33]. Entering into marriage and entering into cohabitation may be associated with reduced alcohol intake in men and women in their late twenties and early thirties [23]. The present study suggests that cohabiting as a couple is associated with reduced risk of problematic drinking in men and women in their thirties and forties. Alcohol drinking and low mood often coexist and more longitudinal evidence is required to determine whether drinking causes low mood or vice versa [24]. The present longitudinal study suggests that a high malaise score is a risk factor for problematic drinking. A longitudinal study in the US suggests that the relationship may be bidirectional: men and women who drink to alleviate low mood are at increased risk of developing alcohol dependency [34]. Data from prospective studies in Australia and the United States suggest that maternal alcohol use is associated with problematic drinking in 21-year-old adults [25,26]. The present study suggests that maternal alcohol use is also associated with problematic drinking in men and women in their thirties and forties. Most of the research about risk factors for problematic drinking is cross-sectional and more longitudinal research is required, including longitudinal studies with ambulatory assessments of alcohol drinking and risk factors for problematic drinking [6].

A total of 966,040 adults have been in contact with drug and alcohol treatment services in England since records began in 2005 and by 31 March 2019, 15% were still engaged in treatment, 40% had left and not completed their treatment and not returned, and 45% had completed their treatment and not returned [2]. It is important to have identified risk factors for problematic drinking in the present study because existing interventions have had limited success [35]. School-based interventions may increase knowledge and improve attitudes towards drinking, but have little impact on behavior in the long term [35]. Motivational interviewing and other brief interventions may reduce drinking in patients who screen positive for harmful drinking but are not alcohol dependent, but have limited effectiveness in adults with more severe alcohol problems [35]. Increasing the price of alcohol is effective in reducing consumption [35], but punitive policies are controversial in liberal societies [36].

The present study has some limitations. Some variables were self-reported and are subject to biases. Different questionnaires were used to assess problematic drinking in participants’ thirties and forties, which may introduce bias. Primary care data suggest that the 4-item CAGE questionnaire has a sensitivity of 71% and a specificity of 91%, while the 10-item AUDIT questionnaire has a sensitivity of 62% and a specificity of 92% [17,37]. When screening algorithms were applied to 1000 hypothetical primary care attendees, the overall accuracy of the CAGE questionnaire was 86% and the overall accuracy of the AUDIT questionnaire was 87% [17]. While the CAGE and AUDIT_PC questionnaires are valid screening tools, they are not clinical diagnoses of alcohol use disorder. There were some missing data; however, there is no need to have a complete dataset when applying multilevel model analysis to longitudinal data because multilevel model analysis is able to cope with missing data [28]. Indeed, it has been shown that applying multilevel analysis to an incomplete dataset is better than applying imputation methods [38,39]. The 1970 British Cohort Study may be representative of those born at the time in England, Scotland and Wales [7], but this sub-sample may not be representative. Cohort members who were included in the present analysis may have been healthier and better educated than those who were not included; however, the frequency of problematic drinking was similar in cohort members who were and were not included in the present analysis.

## 5. Conclusions

This longitudinal study with repeated measurements helps clarify complex associations between potential risk factors and problematic drinking in men and women in their thirties and forties. Formerly smoking, currently smoking, having a degree, having malaise, and having a mother who drank while pregnant may be risk factors for problematic drinking in men. Formerly smoking, currently smoking, being physically active in one’s leisure time, having a degree, having a managerial or professional occupation, having malaise, and having a mother who drank while pregnant may be risk factors in women. Cohabiting may be a negative risk factor for problematic drinking in men and women.

## Figures and Tables

**Table 1 ijerph-19-10664-t001:** Male participants’ characteristics.

Variable	Wave			
	Age 30 (n = 3340)	Age 34 (n = 2658)	Age 42 (n = 2532)	Age 46 (n = 1549)
Problematic drinking, No. (%)				
No	2703 (80.93)	2065 (77.69)	1682 (66.43)	1058 (68.30)
Yes	637 (19.07)	593 (22.31)	850 (33.57)	491 (31.70)
Smoking, No. (%)				
Never smoked	1472 (44.07)	1204 (45.30)	1232 (48.66)	782 (50.48)
Former smoker	627 (18.77)	621 (23.36)	694 (27.41)	507 (32.73)
Current smoker	1241 (37.16)	833 (31.34)	606 (23.93)	260 (16.79)
Leisure-time physical activity, No. (%)				
None	664 (19.88)	570 (21.44)	567 (22.39)	295 (19.04)
Once a week or less	938 (28.08)	739 (27.80)	290 (11.45)	148 (9.55)
Two or three times a week	912 (27.31)	739 (27.80)	743 (29.34)	415 (26.79)
Four or more times a week	826 (24.73)	610 (22.95)	932 (36.81)	691 (44.61)
Highest academic qualification, No. (%)				
None	857 (25.66)	641 (24.12)	686 (27.09)	310 (20.01)
GCSE or equivalent	1301 (38.95)	989 (37.21)	803 (31.71)	548 (35.38)
A-level or equivalent	409 (12.25)	349 (13.13)	329 (12.99)	224 (14.46)
Degree or higher degree	773 (23.14)	679 (25.55)	714 (28.20)	467 (30.15)
Occupation, No. (%)				
Unskilled, semi-skilled or skilled	1908 (57.13)	1418 (53.35)	1173 (46.33)	672 (43.38)
Managerial or professional	1432 (42.87)	1240 (46.65)	1359 (53.67)	877 (56.62)
Cohabiting as couple, No. (%)				
No	1122 (33.59)	620 (23.33)	489 (19.31)	278 (17.95)
Yes	2218 (66.41)	2038 (76.67)	2043 (80.69)	1271 (82.05)
Malaise score				
Low, No. (%)	3067 (91.83)	2403 (90.41)	2215 (87.48)	1356 (87.54)
High, No. (%)	273 (8.17)	255 (9.59)	317 (12.52)	193 (12.46)
Mother drank during pregnancy, No. (%)				
No	1660 (49.70)	1308 (49.21)	1232 (48.66)	738 (47.64)
Yes	1680 (50.30)	1350 (50.79)	1300 (51.34)	811 (52.36)
Father’s occupation in 1970, No. (%)				
Unskilled, semi-skilled or skilled	2271 (67.99)	1780 (66.97)	1668 (65.88)	986 (63.65)
Managerial or professional	1069 (32.01)	878 (33.03)	864 (34.12)	563 (36.35)

GCSE is general certificate of education, a qualification usually sought around 16 years of age. A-level is advance level, a qualification usually sought around 18 years of age.

**Table 2 ijerph-19-10664-t002:** Female participants’ characteristics.

Variable	Wave			
	Age 30 (n = 2915)	Age 34 (n = 2326)	Age 42 (n = 2474)	Age 46 (n = 1526)
Problematic drinking, No. (%)				
No	2679 (91.90)	2005 (86.20)	2082 (84.16)	1281 (83.94)
Yes	236 (8.10)	321 (13.80)	392 (15.84)	245 (16.06)
Smoking, No. (%)				
Never smoked	1436 (49.26)	1146 (49.27)	1274 (51.50)	816 (53.47)
Former smoker	560 (19.21)	572 (24.59)	707 (28.58)	461 (30.21)
Current smoker	919 (31.53)	608 (26.14)	493 (19.93)	249 (16.32)
Leisure time physical activity, No. (%)				
None	622 (21.34)	465 (19.99)	736 (29.75)	386 (25.29)
Once a week or less	812 (27.86)	550 (23.65)	331 (13.38)	158 (10.35)
Two or three times a week	710 (24.36)	632 (27.17)	755 (30.52)	444 (29.10)
Four or more times a week	771 (26.45)	679 (29.19)	652 (26.35)	538 (35.26)
Highest academic qualification, No. (%)				
None	563 (19.31)	415 (17.84)	510 (20.61)	233 (15.27)
GCSE or equivalent	1184 (40.62)	884 (38.01)	840 (33.95)	537 (35.19)
A-level or equivalent	451 (15.47)	406 (17.45)	406 (16.41)	275 (18.02)
Degree or higher degree	717 (24.60)	621 (26.70)	718 (29.02)	481 (31.52)
Occupation, No. (%)				
Unskilled, semi-skilled or skilled	1733 (59.45)	1217 (52.32)	1234 (49.88)	753 (49.34)
Managerial or professional	1182 (40.55)	1109 (47.68)	1240 (50.12)	773 (50.66)
Cohabiting as couple, No. (%)				
No	843 (28.92)	550 (23.65)	543 (21.95)	328 (21.49)
Yes	2072 (71.08)	1776 (76.35)	1931 (78.05)	1198 (78.51)
Malaise score				
Low, No. (%)	2579 (88.47)	1957 (84.14)	2038 (82.38)	1259 (82.50)
High, No. (%)	336 (11.53)	369 (15.86)	436 (17.62)	267 (17.50)
Mother drank during pregnancy, No. (%)				
No	1370 (47.00)	1091 (46.90)	1158 (46.81)	696 (45.61)
Yes	1545 (53.00)	1235 (53.10)	1316 (53.19)	830 (54.39)
Father’s occupation in 1970, No. (%)				
Unskilled, semi-skilled or skilled	1927 (66.11)	1536 (66.04)	1620 (65.48)	962 (63.04)
Managerial or professional	988 (33.89)	790 (33.96)	854 (34.52)	564 (36.96)

GCSE is general certificate of education, a qualification usually sought around 16 years of age. A-level is advance level, a qualification usually sought around 18 years of age.

**Table 3 ijerph-19-10664-t003:** Longitudinal associations of modifiable and non-modifiable risk factors with problematic drinking in men.

Potential Risk Factor	Odds Ratio (95% Confidence Interval)
Smoking	
Never smoked	Reference
Former smoker	3.06 (2.52, 3.72)
Current smoker	3.76 (3.09, 4.57)
Leisure-time physical activity	
None	Reference
Once a week or less	1.10 (0.90, 1.35)
Two or three times a week	1.02 (0.84, 1.24)
Four or more times a week	1.06 (0.88, 1.28)
Highest academic qualification	
None	Reference
GCSE or equivalent	0.92 (0.74, 1.14)
A-level or equivalent	0.91 (0.68, 1.21)
Degree or higher degree	1.31 (1.02, 1.71)
Occupation	
Unskilled, semi-skilled or skilled	Reference
Managerial or professional	0.97 (0.83, 1.15)
Cohabiting as a couple	
No	Reference
Yes	0.72 (0.62, 0.85)
Malaise score	
Low	Reference
High	1.78 (1.44, 2.21)
Mother drank during pregnancy	
No	Reference
Yes	1.70 (1.43, 2.02)
Father’s occupation in 1970	
Unskilled, semi-skilled or skilled	Reference
Managerial or professional	1.01 (0.83, 1.22)

**Table 4 ijerph-19-10664-t004:** Longitudinal associations of modifiable and non-modifiable risk factors with problematic drinking in women.

Potential Risk Factor	Odds Ratio (95% Confidence Interval)
Smoking	
Never smoked	Reference
Former smoker	2.74 (2.16, 3.48)
Current smoker	4.92 (3.83, 6.31)
Leisure-time physical activity	
None	Reference
Once a week or less	1.06 (0.81, 1.38)
Two or three times a week	1.32 (1.04, 1.68)
Four or more times a week	1.31 (1.03, 1.66)
Highest academic qualification	
None	Reference
GCSE or equivalent	1.30 (0.97, 1.74)
A-level or equivalent	1.29 (0.91, 1.83)
Degree or higher degree	2.01 (1.43, 2.81)
Occupation	
Unskilled, semi-skilled or skilled	Reference
Managerial or professional	1.40 (1.15, 1.70)
Cohabiting as a couple	
No	Reference
Yes	0.73 (0.60, 0.90)
Malaise score	
Low	Reference
High	1.95 (1.56, 2.43)
Mother drank during pregnancy	
No	Reference
Yes	1.44 (1.17, 1.78)
Father’s occupation in 1970	
Unskilled, semi-skilled or skilled	Reference
Managerial or professional	0.86 (0.68, 1.08)

## Data Availability

Registered users can access the data used in the present study via the UK Data Service: www.ukdataservice.ac.uk. The analysis plan is available from the corresponding author.

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
