# Peer review of "Risk Factors for Problematic Drinking in One’s Thirties and Forties: A Longitudinal Analysis of the 1970 British Cohort Study"

_ijerph, 2022, doi:10.3390/ijerph191710664_

Round 1
Reviewer 1 Report
This is an interesting study which explored the risk factors of problematic drinking in a British Cohort Study of multiple waves. It is an important topic and the study dataset presents an exciting opportunity to understand the risk factors of problematic drinking. However, the paper would be improved by further clarification of the statistical method and study design. This discussion appeared to be more of a pure comparison with findings with previous studies, without enough interpretation of the results and discussion of the implications and potential mechanisms of the findings.
1) L35: what does “control of alcohol drinking” refer to here?
2) Materials and methods, participants: please provide more details of the study population of the 1970 British Cohort Study, including the number of participants, response rates and loss to follow-up for each survey, any inclusion criteria into the 1970 British Cohort Study and for this particular paper.
3) L60-61 and Institutional Review Board Statement: It is strange to report a particular ethical approval (the age 46 survey) as an example. Please include all relevant ethical approval references for all surveys used.
4) L92-93: “The postal questionnaire was not received from 1,107 cohort members at age 42.” What was the total number of cohort members being sent the postal questionnaire? It is important to report the response rate and sample size for each survey, especially when the interview method and questionnaire differs hugely between each survey which might introduce biases (e.g. sampling bias, survivor bias, reporting bias.
5) Statistical methods: what covariates were adjusted in the model?
6) Statistical methods: I suppose due to varying response rates and various reason, there would inevitably missing data on paternal occupation and maternal drinking history for the participants at age 30-40s. How were missing data handled (I recognised it was mentioned towards the end of the paper but it would be helpful to be addressed upfront)? This also linked back to the need to have a clear flowchart of participants included through different surveys.
7) L139-140 and Supplementary Table 1: There were almost half of the total study population not being included in the present analysis, what were the reasons for them not being included?
8) Table 1&2: There was an increase in proportion of people having a degree or higher degree across wave. Presumably most of the participants involved in different waves were the same people, and that most of them would have obtained a degree by the time of the first wave (age 30), the highest educational level should remain mostly constant over time. How would the authors explain the increase in proportion of people having a degree? Would it suggest survival bias (as those with lower education and lower SES would have experienced ill health and potentially died early thus no longer be able to participate in later waves)?
9) L154: Should be Table 2 for women instead of Table 1.
10) Tables 3-4 and Supplementary Tables 2-3: It would be helpful to report the number of cases and non-cases in these tables so readers can judge the power of the analyses, and how much reduction in the sample size was between Table 3/4 and Supplementary Table 2/3. That might give some indications on why the association with those having higher education disappeared in sensitivity analysis.
11) L172 -183, L194-205: It would be helpful to at least report some of the key ORs (95% CIs) in the text.
12) L252-255: I was not sure if the present study suggest that women who are more physically active are more susceptible to problem drinking, as the associations were no longer there in sensitivity analysis. It would be helpful to see a more specific discussion on this. Also what would be the potential explanation for the differences by sex?
13) L257-260: Similar to findings of physical activity, the higher odds of problem drinking associated with managerial or professional occupation was only observed in women but not men. What would be the potential explanation for this higher risks and sex difference?
14) L264-267: It was not clear enough to me who the previous study on using alcohol to alleviate low mood was linked to (or if it was supposed to be linked to) the current findings of higher malaise score relating to problem drinking. More explicit discussion of the authors’ interpretation of the findings is needed.
15) L271-273: The statement on future research seems to be disconnected from the previous sentence and vague (e.g. what kind of research questions to be addressed and what kind of data should future longitudinal studies include?). It would be helpful to add a bridging sentence to discuss what important questions the present study was yet to answer, and then introduce idea for future research.
16) The authors briefly touched on the point of using different questionnaires to assess problem drinking at different waves. This is an important limitation and an expanded discussion is needed (e.g. sensitivity of different questionnaires? Severity of problem drinking detected by different questionnaires?)
17) Several other discussion points as potential limitations of the study, including survivor bias especially given the longitudinal study design, temporality and directionality of the associations.
Reviewer 2 Report
The paper is an interesting review of some risk factors related to problematic alcohol use in long-term follow-up.
However, some issues are problematic and need to be explained by the authors.
Introduction:
- the rationale for variation in risk factors across the life course is very simple and does not really allow for understanding change in the influence of risk factors.
- The explanation of the British Cohort study is very simple in the introduction and adds nothing to the introduction.
- The wording of the objective is poor and unclear.
Dependent variables: The use of different questionnaires to measure problematic use is unclear as each measures different aspects of problematic use. In my opinion it is a mistake to first use a screening questionnaire for alcoholism and then use a questionnaire for problem drinking. This approach is a mistake in the longitudinal conception of the addictive process.
Independent variables: It is not explained why these variables are selected and not others. This choice is not justified by the use of any theoretical reference model.
Results: the results are adequate, although they are contaminated by the problems mentioned above.
Discussion:
- The future implications of the results of this work are not adequately indicated. Furthermore, it is only pointed out that it may have implications for PRIMARY PREVENTION, when in the framework of prevention work this classification was abandoned many years ago and the classification proposed by Gordon (1987) is used. It would be necessary to redo this part, clearly indicating the possible applications of the work.
- From line 277 to line 285 a number of statements are made about the ineffectiveness of prevention that are poorly based and poorly connected to each other. These statements not only contribute nothing as they are written in the paper, but even contradict each other. This whole point needs to be reworked.
Round 2
Reviewer 2 Report
Thank you for the review and the changes made.